# Notifications on Pesticide Residues in the Rapid Alert System for Food and Feed (RASFF)

**DOI:** 10.3390/ijerph19148525

**Published:** 2022-07-12

**Authors:** Marcin Pigłowski

**Affiliations:** Department of Quality Management, Faculty of Management and Quality Science, Gdynia Maritime University, Morska 81–87 Str., 81-225 Gdynia, Poland; m.piglowski@wznj.umg.edu.pl

**Keywords:** cluster analysis, European Union, food safety, pesticide residues, RASFF

## Abstract

Pesticides are commonly used to protect plants against various pests and to preserve crops, but their residues can be harmful for human health. They are the third most widely reported hazard category in the European Rapid Alert System for Food and Feed (RASFF). The purpose of the study was to identify the most frequently notified pesticides in the RASFF in 1981–2020, considering: year, notification type, product category, origin country, notifying country, notification basis, distribution status and action taken. The data from the RASFF database was processed using: filtering, transposition, pivot tables and then subjected to cluster analysis: joining (tree clustering) and two-way joining methods. Pesticides were most commonly reported in fruits and vegetables and herbs and spices following border controls and rejections. The products usually came from India or Turkey and were not placed on the market or were not distributed and then destroyed. The effectiveness of the European Union border posts in terms of hazards detection and mutual information is important from the point of view of protecting the internal market and ensuring public health. It is also necessary to increase the awareness of pesticide users through training and the activity of control authorities in the use of pesticides.

## 1. Introduction

Pesticides are chemicals designed to protect food by controlling harmful insects, diseases, rodents, weeds, bacteria and other pests [1,2]. They can destroy, suppress or alter the life cycle of pests [3]. In a broader context, pesticides are therefore used to: kill, repel or control pests to protect crops before and after harvest, influence the life process of plants, destroy weeds and prevent their growth, and preserve plant products [4]. It is estimated that if pesticides were not used, a third of crops would be lost [2]. However, if pesticides are used in an improper way, they can also be harmful to people, animals and the environment [1].

Regulation (EC) 396/2005 on maximum residue levels (i.e., MRLs) of pesticides in or on food and feed of plant and animal origin defines pesticide residues as residues, including active substances, their metabolites and/or breakdown or reaction products of active substances currently or formerly used in plant protection products, which are present in or on products, including, in particular, those which may arise as a result of use in plant protection, in veterinary medicine and as a biocide. The MRL means the upper legal level of concentration of pesticide residues in or on food or feed established in accordance with this regulation, based on good agricultural practice and the lowest consumer exposure necessary to protect vulnerable consumers [5]. Under Regulation (EC) 396/2005, the European Food Safety Authority (EFSA) provides annual reports, which examine pesticide residue levels in food on the European Union (EU) market. The main results of these reports from the last five years available (2016–2020) are shown in Table 1. These results are based on the analysis of an average of around 90,000 product samples from both EU and non-EU countries each year [6,7,8,9,10].

The results published by the EFSA indicate that about 95% of the samples are within the legal limits, of which about 40% did not even exceed the limit of quantification (LOQ), and only in 2–3% of the samples (taking into account the measurement uncertainty) were the MRLs exceeded. In the aforementioned reports, the EFSA points out that the probability of European citizens being exposed to levels of pesticide residues that could lead to adverse health effects is low. However, it consistently recommends the need to improve the efficiency of the European control system for pesticide residues [6,7,8,9,10], including optimising traceability [8].

Due to the widespread use of pesticides, they can come into contact with humans through various routes [4]. Therefore, pesticide residues may be present not only in plants, but also in meat, milk, eggs as a result of farm animal consumption, feed from treated crops, environmental contamination and spray drift [2]. Aggregate exposure may also occur for some pesticides if, for example, their residues are present in both food and insect baits [3].

Furthermore, climatic conditions in some countries can promote plant diseases (caused by plant pathogens), plant pests (plant-feeding or -sucking insects and mites) and weeds, and thus reduce yields [11]. In this context, it is therefore very important to apply good agricultural practice (GAP), which means the nationally recommended, authorised or registered safe use of plant protection products under actual conditions at any stage of production, storage, transport, distribution and processing of food. In addition, the principles of integrated pest control in a given climate zone should be implemented, as well as the use of a minimum quantity of pesticides and the setting of MRLs/temporary MRLs at the lowest level that achieves the intended effects. However, in certain cases it is necessary to implement a critical GAP, which means more than one GAP for an active substance/product combination, leading to an increase in the highest acceptable level of pesticide residues in treated crops and the basis for establishing a MRL [5].

The EU Pesticide Database covers 1472 active substances, safeners and synergists, of which 452 are approved and 937 are not approved, whereas 64 are under evaluation. This database also includes information on pesticide residues and the MRLs that apply for such residues in food products—there are 654 records (as of 5.07.2022) [12]. Active substances that are not approved are considered obsolete, but their stockpiles are still held by third countries from where the food is imported [3]. The problem with the presence of pesticide residues in food and feed is also noticeable in the Rapid Alert System for Food and Feed (RASFF) (Figure 1) [13]. The notification on pesticide residues in the RASFF is made when a substance in question was not approved or an MRL was exceeded (but for some notifications, values are not given or several values are given, making potential analysis problematic). In 2015, the number of notifications continued to decline slightly, which may be explained by the fact that the EU entry points strengthened border checks [14]. However, in 2020 the number of these notifications increased significantly.

It is also very important to note that the hazard category “pesticide residues” is the third most frequently reported in the RASFF, lead only by pathogenic micro-organisms and mycotoxins (Table 2).

The RASFF is a tool that ensures a rapid flow of information, which enables a quick response when a risk to public health is detected in the food chain. It was created as early as 1979, but its legal basis is now the Regulation (EC) 178/2002, which lays down the general principles and requirements of food law, establishes the European Food Safety Authority (EFSA) and lays down procedures in matters of food safety [15]. Under this regulation, RASFF members are obliged to immediately notify the European Commission of any information concerning a serious health risk deriving from food or feed, and the commission transmits this information to other members of the system [16]. The members of the RASFF are the 27 countries of the EU, the European Commission, the EFSA, the European Free Trade Association Surveillance Authority (ESA), and Norway, Liechtenstein, Iceland and Switzerland.

In the RASFF, alert notifications are sent when food or feed presenting a serious health risk is already on the market and rapid action is needed (e.g., product withdrawal). A RASFF member passes on information within the system so that others can check whether the product is also in their market and take appropriate measures accordingly. Information notifications are used when a risk concerning food or feed has been identified but other RASFF members do not need to take rapid action. This is because the product may not have reached their market yet, is no longer present in their market or the nature of the risk does not require rapid action. Border rejections concern consignments of food or feed that have been tested and rejected at the external border of the European Economic Area (EEA) if a health risk has been identified. Notifications are sent to other EEA border posts to ensure that the rejected consignment does not re-enter through another border post [15].

However, in publications the problem related to pesticide residues in food and feed is addressed usually in a generalised, selective way and in a short time horizon. An exception here is the extensive analysis on pesticide residues in RASFF notifications from 2002 to 2020 recently carried out by Kuchheuser and Birringer (relating also to MRLs) [17,18], but the individual issues (variables) can be presented in an even more comprehensive and cross-sectional way. Pesticide notifications are also signalled in the annual reports of the RASFF or the EFSA, but only apply to the year for which the report is issued. Therefore, the purpose of the study was to identify the most frequently notified pesticides in the RASFF between 1981 and 2020, taking into account: year, notification type, product category, origin country, notifying country, notification basis, distribution status and action taken.

## 2. Materials and Methods

### 2.1. Data and Its Processing

The current ongoing RASFF database, maintained by the European Commission, provides only data up to two years back, i.e., it does not include historical data [19]. Therefore, the data was extracted from the RASFF notifications pre-2021 public information database and covered 8013 notifications on pesticide residues between 1981 and 2020 (a 40-year period) [13]. These notifications were very diverse, as they concerned 257 different pesticides. Due to this wide variability, all pesticides were included only in preliminary studies (without indication of individual substances). However, detailed studies (i.e., with indication of the pesticide concerned) were carried out on only the first seventeen most frequently notified pesticides, accounting for more than half of all notifications. They covered 4061 notifications in the period 1994–2020, i.e., 27 years. All pesticides with the number of notifications were collected in Appendix A in Appendix A.

The data from the RASFF pre-2021 database were processed in Microsoft Excel using: filtering, transposition, pivot tables and vertical lookup. Finally, the following eight variables were adopted in the study: year, notification type, product category, origin country, notifying country, notification basis, distribution status and action taken. In the case of the variable “notification type” three values were assumed: alert, border rejection and information as a combination of several values, i.e., information in the period 1994–2010, information for attention (from 2011) and information for follow-up (also from 2011). In turn, in the case of the variables origin country, notification basis, distribution status and action cells without values (empty cells), they were completed by the phrase “(not specified)”. To make the figures more readable, in the case of the variables product category, notification basis, distribution status and action taken, the names of some values were shortened (Appendix A in Appendix A).

### 2.2. The Cluster Analysis

The data were then transferred to source tables in Statistica 13.3 (TIBCO Software Inc., Palo Alto, CA, USA), where they were subjected to cluster analysis by joining (tree clustering) and two-way joining methods. For the purpose of cluster analysis, years where no notifications were reported were filled in with the value “0”. In the case of the joining method, in the source tables, pesticides were in rows and values of eight variables (year, notification type, product category, origin country, notifying country, notification basis, distribution status and action taken) were in columns. The following settings were used: Euclidean distance for the distance measure and Ward’s method for amalgamation. The Euclidean distance is the geometric distance in a multidimensional space. Ward’s method uses an analysis of variance to evaluate the distances between clusters, attempting to minimise the sum of squares of any two (hypothetical) clusters, which can be formed at each step. This method is considered to be very efficient; however, it tends to create clusters of small sizes (flattening of clusters). The results of the joining cluster analysis were presented as vertical icicle plots in Appendix A (8 charts in total) in Appendix A.

Then, the source tables for the two-way joining cluster analysis were prepared. For each of the seventeen pesticide studied, seven tables were built, with years in columns and the values of the other variables (i.e., notification type, product category, origin country, notifying country, notification basis, distribution status and action taken) in rows. This method was used to discover cluster patterns that may appear simultaneously in particular years with given values of other variables. Although clusters may not be homogeneous in nature, this method is considered a useful analysis tool [20].

The results of the two-way joining cluster analysis were presented in Appendix A (with particular charts) in Appendix A (119 charts in total). The clusters in the two-way joining cluster analysis were shown by coloured squares (from green, through yellow, orange and red, to brown) in contour/discrete charts. In order to increase the readability of the figures, the dark green colour (smallest clusters or no clusters) was changed to white. During the presentation of the results, attention was paid only to the largest clusters (yellow, orange, red and brown colours).

The years with the highest number of notifications for a given pesticide were determined by the variable “notification basis”. Due to the large diversity in the values of the variable “origin country”, in the case of some pesticides only the first 30 countries with the highest number of notifications were taken into account. If, for individual values (within each variable), notifications were not reported in a similar way, only years with the highest number of notifications were indicated. When preparing the charts, it was also necessary to rearrange the data matrix for some variables. Moreover, some results were discarded in order to maintain the most accurate coverage of the values in individual years for different variables.

## 3. Results

### 3.1. General Results

#### 3.1.1. Percentage Share of Notifications

The number and percentage of notifications on pesticide residues in the RASFF, depending on the particular variables, are presented in Figure 2. In the case of notification type, all three types were presented. For the variables of product category, origin country, notifying country, notification basis, distribution status and action taken, values with notifications above the mean (from largest to smallest) were only shown and notifications for the remaining values were summarised and named “others”.

The border rejections accounted for nearly half of all notifications (48.4%), information notifications for 36.6% and alerts only for 15.0% (Figure 2a). Notifications were related mainly to fruits and vegetables (67.6%), as well as herbs and spices (10.0%), cocoa and cocoa preparations, coffee and tea (7.8%) and nuts, nut products and seeds (5.3%) (Figure 2b). The reported products originated mainly from India (18.1%) and Turkey (17.6%), as well as from China (7.8%), Thailand (6.2%), Egypt (5.8%) and Italy (3.7%) (Figure 2c). They were notified by Germany (13.3%), Bulgaria (12.1%), the United Kingdom (10.2%), the Netherlands (9.1%), France (8.2%) and Italy (8.0%) (Figure 2d).

The notifications were based on border control, after which the consignment was detained (47.2%), as well as official controls in the market (26.4%) and the company’s own checks (10.9%) (Figure 2e). Products were not placed on the market (29.0%), were not distributed (18.6%) or were distributed to other countries (13.2%) (Figure 2f) and the most common actions taken against them were destruction (26.7%), withdrawal from the market (9.9%) and re-dispatch (8.9%) (Figure 2g).

#### 3.1.2. Results of Joining Cluster Analysis

The results presented in Figure 1 and Figure 2 were confirmed by the cluster analysis using the joining method (Appendix A in Appendix A). This analysis showed a separate cluster for 2020, when there was a sharp increase in the number of notifications, whereas the years 2002 and 2007–2019 formed a second cluster. Here, however, it can also be further noted that the notifications in 2008–2009, 2012–2013, 2014–2015 and 2018–2019 were very similar in nature (Appendix A). It should also be pointed out that information notifications and border rejections were also similar during the period considered (Appendix A). If the product category is taken into account, a clearly separate cluster was created by fruits and vegetables due to the significant number of notifications regarding these products. Next to them, however, there were also nuts, nut products and seeds, herbs and spices, cocoa and cocoa preparations, and coffee and tea, as well as eggs and egg products, and cereals and bakery products (Appendix A). Notifications mostly concerned products from India and Turkey, which is why these countries created a separate cluster (Appendix A)

The countries that made the largest number of notifications (i.e., Bulgaria, the Netherlands, the United Kingdom, Germany, France, Italy and Belgium) also formed a separate cluster (Appendix A). Taking into account the notification basis, the border controls (after which the consignment was detained), the official controls in the market and the company’s own checks were of a similar nature (Appendix A). In the case of distribution status, the notifications were similar for products that were not placed on the market and those that were not distributed (Appendix A). The results of the cluster analysis using the joining method with regard to the actions taken were very varied; however, it can be seen that an action such as destruction is the first element of a separate cluster (Appendix A).

### 3.2. Results of Two Way-Joining Cluster Analysis for Pesticides Studied

Results of the two-way joining cluster analysis related to notifications on pesticides in the RASFF are presented in Table 3. In this table the number of notifications on particular pesticide and the number of the figure in the Appendix A are also given. If there was a particularly high number of notifications for a particular product, the name of that product was also given (this data came directly from the RASFF pre-2021 database) [13].

#### 3.2.1. Fruits, Vegetables and Nuts from India

The most frequently notified were fruits and vegetables from India in 2012–2013. It concerned mainly okra, where the residues of pesticides such as acephate, acetamiprid, dimethoate, methamidophos, monocrotophos, profenofos and triazophos were found. The notification basis for these products was border control, after which the consignment was detained, resulting in border rejection. These products were notified by the United Kingdom and France, they were not distributed and the most common action taken against them was destruction. The notifications related to profenofos and triazophos also applied to curry (product category “herbs and spices”).

However, recently (in 2020), the Netherlands reported a very significant number of notifications for sesame seeds (product category “nuts, nut product and seeds”) from India where residues of ethylene oxide were found. These products were already on the European Union market because the notification basis was the company’s own check, and the notification type was an alert. They were also already distributed to other member countries; therefore, consignors and recipients were informed, and products were withdrawn from the market and even recalled from consumers. It is also worth noting that residues of another pesticide (carbendazim) on fruits and vegetables from India were notified by Italy in 2014–2015.

#### 3.2.2. Fruits and Vegetables from Turkey

Bulgaria notified fruits and vegetables from Turkey. In this category, residues of acetamiprid on peppers and pomegranates (in 2012–2013 and 2020), chlorpyrifos on lemons, peppers and stuffed vine leaves (in 2015–2020) and formetanate, methomyl and oxamyl on peppers (in different years) were found. Methomyl was also reported by Germany, the Netherlands and France and oxamyl by Germany. The notification basis in this case was usually border control and the consignment was detained, and therefore it was rejected at the border. The reported peppers were not distributed and they were destroyed.

#### 3.2.3. Vegetables from Nigeria

Dichlorvos was notified by the United Kingdom on beans (product category “fruits and vegetables”) from Nigeria in 2013–2015. Similarly, the notification basis was border control and the consignment was detained (the notification type was border rejection). The product was not placed on the market and the actions taken with regard to it consisted of official detention, re-dispatch or destruction.

#### 3.2.4. Eggs from Italy

The last pesticide to look out for is fipronil on eggs and egg products. It was reported by Italy in 2017 on products originating from that country. The notification basis was official controls in the market and the reporting was carried out under information notifications. The distribution was restricted to the notifying country and the product was withdrawn from the market.

#### 3.2.5. Other Notifications

Notifications related to carbofuran (2014–2019) and omethoate (2005–2013, 2018–2019) were very diverse; however, they also concerned fruits and vegetables. These products with residues of the above mentioned pesticides usually originated from Asia (mainly Thailand) and notifications were reported by the Western European countries. It is also noteworthy that when comparing the results presented in Figure 2 and Table 3, some differences with respect to some values can be seen. The most visible is the lack of the product category “cocoa and cocoa preparations, coffee and tea” in Table 3, which means that pesticides other than those most frequently reported were notified on these products.

## 4. Discussion

Some authors only signalled the reporting of pesticide residue hazards in the RASFF in particular years: 2004 [21], 2004–2014 [22], 2007–2013 [23], 2008 [24], before 2012 [25] and 2014 [26]. In turn, other authors indicated period, product (product category) and additional information (Table 4). Some articles pointed out only particular pesticides without indication of the products: acephate, methamidophos and monocrotophos (in 2002–2003) [27], acetamiprid and imidacloprid (in 2012–2015) [28], acetamiprid, carbendazim, carbofuran, chlorpyrifos, dichlorvos, dimethoate, formetanate and other pesticides (in 2015, notified mainly by Italy, France, Belgium and the Netherlands) [14], isofenphos-methyl and omethoate (in 2015) [2], chlorpyrifos (in 2016–2017) [29], and also boscalid, chlorpyrifos and tebuconazole in peaches and nectarines from Turkey [30] or carbendazim, cypermethrin, dimethoate, endosulfan and ethion [31].

It is worth noting that the most frequently indicated were those pesticides for which a detailed analysis of notifications in the RASFF was performed (Table 3). Fruits and vegetables and herbs and spices were the most commonly identified (this was also pointed out in Figure 2b), but the presence of pesticide residues in okra and curry leaves from India seems to be a frequently noticed problem. In fact, these products (and others) were covered by three legislations (now no longer in force): Commission Regulation (EC) 669/2009 implementing Regulation (EC) 882/2004 of the European Parliament and of the Council as regards the increased level of official controls on imports of certain feed and food of non-animal origin, Commission Implementing Regulation (EU) 91/2013 laying down specific conditions applicable to the import of groundnuts from Ghana and India, okra and curry leaves from India, and watermelon seeds from Nigeria, and then, Commission Implementing Regulation (EU) 885/2014 laying down specific conditions applicable to the import of okra and curry leaves from India. The adoption of Regulation 91/2013 was the result of an audit that was conducted in India in 2011 by the Food and Veterinary Office. At that time, it was found that there was no framework for the use of pesticides for okra, curry leaves and other products, and the Indian authorities were unable to provide the European Commission with a satisfactory action plan to address the identified shortcomings [32]. Currently, in the scope of checks for the presence of pesticide residues in okra and curry leaves from India, Commission Implementing Regulation (EU) 2019/1793 on the temporary increase in official controls and emergency measures governing the entry into the Union of certain goods from certain third countries is in force (it also applies to other products, hazards and origin countries) [33], but this regulation has also already been changed.

In recent years, however, the two biggest pesticide crises have involved other product categories. Fipronil on eggs was first reported by Belgium [55], but the most notifications were submitted by Italy (for products from this country) and there were so many of them that they were recorded among the ten most notified in the 2017 RASFF annual report [56]. Whereas in September 2020, Belgium notified in the RASFF that it had exceeded almost four thousand times the MRL set for ethylene oxide (by Regulation (EC) 396/2005) in sesame seeds imported from India [13]. The Food and Feed Crisis Coordinators meeting was organized immediately [62], and the Commission Implementing Regulation (EU) 2020/1540 amending Implementing Regulation (EU) 2019/1793 on sesame seeds originating from India was issued the following month. This Regulation required that each shipment of sesame seeds from India be accompanied by an official certificate stating that the products had been sampled and analyzed for pesticide residues, and the test results had to be attached to the certificate [63]. The number of ethylene oxide notifications was so high that it was also among the ten most reported in the annual RASFF report for 2020 [59]. In 2021, the number of notifications on pesticide residues remained at a similar high level as in 2020 (i.e., more than 1000). However, the number of notifications relating to the presence of ethylene oxide in nuts (originating from India) decreased several times, while the number of notifications referring to chlorpyrifos in fruits and vegetables increased significantly [19].

## 5. Conclusions

The vast majority of pesticide residue hazards reported in the RASFF in 1981–2020 concerned fruits and vegetables (67.6%), followed by herbs and spices (10.0%), cocoa and cocoa preparations, coffee and tea (7.8%), and nuts, nut products and seeds (5.3%). Notifications were mainly reported on the basis of border controls, as the consignment was detained (47.2%), resulting in border rejections (48.4%). Information notifications and alert notifications were also made, and these were mainly based on official controls in the market and the company’s own checks. The notified products originated mainly from India (18.1%) and Turkey (17.6%), as well as China, Thailand, Egypt and Italy, and were reported by Germany, Bulgaria, the United Kingdom, the Netherlands, France, Italy and Belgium. The reported products were mostly not placed on the market (29.0%), not distributed (18.6%) or distributed to other countries (13.2%), and the following actions were taken against them: destruction (26.7%), withdrawal from the market (9.9%) or re-dispatch (8.9%).

The detailed study covered half of the RASFF notifications for the seventeen most frequently notified pesticides between 1994 and 2020, i.e., a 27-year period. The two-way joining cluster analysis enabled the identification of concentrations and similarities in the notifications indicating, among others: pesticide, years, product, origin country and notifying country. A significant problem was the presence of pesticide residues in okra from India in 2012–2013 notified mainly by the United Kingdom and France (acephate, acetamiprid, dimethoate, methamidophos, monocrotophos, profenofos and triazophos). It is also worth noting that in 2020 there was a very sharp increase in the number of notifications, reported mainly by the Netherlands, due to the presence of ethylene oxide on sesame seeds from India. Notifications reported by Bulgaria on products from Turkey can also be observed (acetamiprid on peppers and pomegranates in 2012–2013 and 2020, chlorpyrifos on lemons, peppers and stuffed vine leaves in 2015–2020, and formetanate, methomyl and oxamyl on peppers in different years). There was also a serious crisis in 2017 when Italy reported fipronil on eggs originating from that country.

The analysis of notifications in the RASFF has shown that the greatest number of hazards of pesticide residues concerned products originating from Asia. Therefore, the effectiveness of the European Union border posts in terms of hazard detection and mutual information transfer is extremely important from the point of view of protecting the internal market and ensuring public health. Important elements of this safety include Regulation (EC) 396/2005 on the maximum residue levels (MRLs) of pesticides, the EU Pesticide Database including active substances, the Regulation (EU) 2019/1793 on the temporary increase in official controls and other regulations as appropriate.

It is necessary to develop cooperation between EU institutions and bodies and their counterparts in non-EU countries. Cooperation could take place at the level of working teams and include advising on the creation or amendment of laws, as well as providing training in these countries. Training should take into account the indigenous or even local nature of production in these countries, be conducted in a structured manner and with the participation of national supervisory authorities and bodies, and cover issues related to the use of pesticides permitted by the EU and that are appropriate for a given plant (season, amount, withdrawal period). This will minimise or avoid image and financial losses (e.g., costs related to transport and disposal of contaminated products) by countries exporting products to the EU market.

Further research on pesticide residues could include linking RASFF notifications to quantities of particular types of imported products (based on Eurostat data) in order to forecast possible hazards. However, this could be hampered by data gaps in the RASFF database (especially for the earlier years of the system operation) and the lack of access to historical data in the currently functioning official database. Therefore, it would be necessary to additionally connect the data from the historical database and the one currently available.

## Figures and Tables

**Figure 1 ijerph-19-08525-f001:**
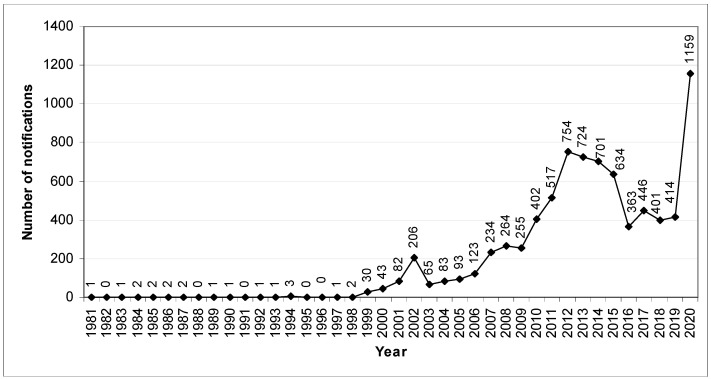
Number of notifications on pesticide residues in the RASFF in 1981–2020.

**Figure 2 ijerph-19-08525-f002:**
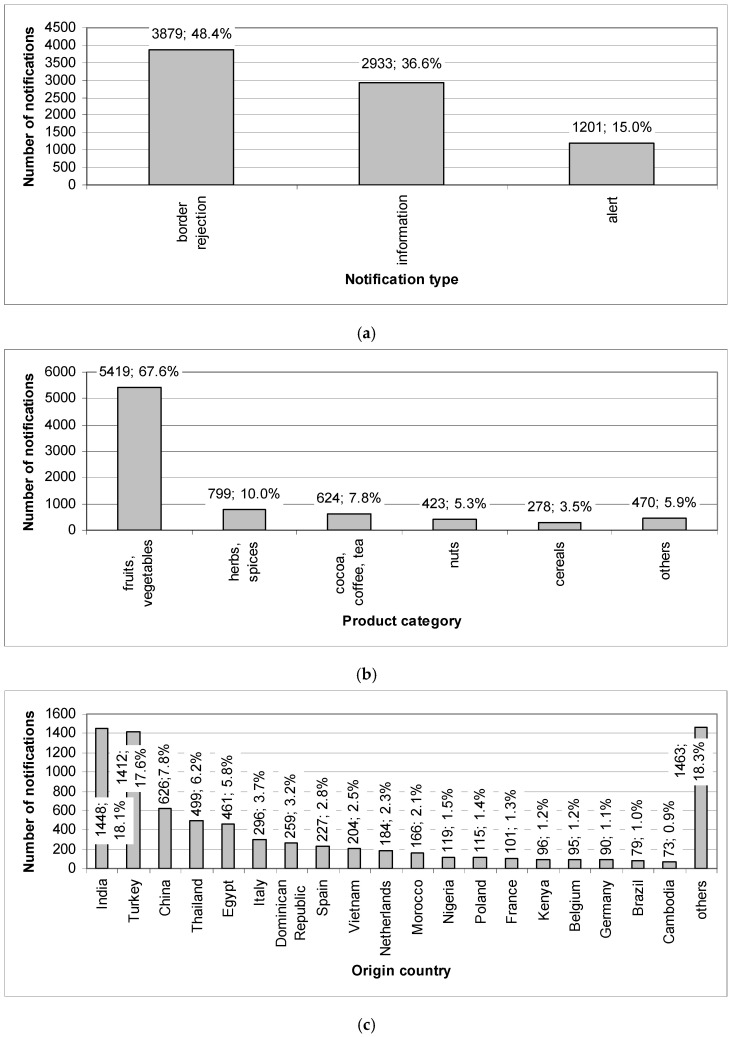
Number and percentage of notifications on pesticide residues in the RASFF by variable.

**Table 1 ijerph-19-08525-t001:** The main results presented in annual reports of the European Food Safety Authority on pesticide residues in food in 2016–2020 (percentage).

Year	Within the Legal Limits	Below the LOQ *	Not Exceeding MRLs **	Exceeding MRLs (***)
2016	96.2%	50.7%	45.5%	3.8% (2.2%)
2017	95.9%	54.1%	41.8%	4.1% (2.5%)
2018	95.5%	NDA	NDA	4.5% (2.7%)
2019	96.1%	NDA	NDA	3.9% (2.3%)
2020	94.9%	54.6%	40.3%	5.1% (3.6%)

* LOQ—limit of quantification, ** MRLs—maximum residue levels, *** taking into account the measurement uncertainty.

**Table 2 ijerph-19-08525-t002:** Number of notifications and percentage in particular hazard categories in the RASFF in 1981–2020.

Hazard Category	Number of Notifications	Percentage	Hazard Category	Number of Notifications	Percentage
Adulteration/fraud	2139	2.8%	Natural toxins (other)	871	1.1%
Allergens	2452	3.2%	Not determined/other	571	0.7%
Biological contaminants (other)	899	1.2%	Novel food	1065	1.4%
Chemical contaminants (other)	27	below 0.05%	Organoleptic aspects	1104	1.4%
Composition	4780	6.3%	Packaging defective/incorrect	502	0.7%
Environmental pollutants	1586	2.1%	Parasitic infestation	822	1.1%
Feed additives	198	0.3%	Pathogenic micro-organisms	14,177	18.6%
Food additives and flavourings	3659	4.8%	Pesticide residues	8013	10.5%
Foreign bodies	2636	3.5%	Poor or insufficient controls	1791	2.3%
Genetically modified food or feed	889	1.2%	Process contaminants	254	0.3%
Industrial contaminants	168	0.2%	Radiation	616	0.8%
Labelling absent/incomplete/incorrect	582	0.8%	Residues of veterinary medicinal products	2697	3.5%
Metals	3585	4.7%	Transmissible spongiform encephalopathy (TSE)	209	0.3%
Microbial contaminants (other)	2591	3.4%			
Migration	4042	5.3%	(Not specified)	27	below 0.05%
Mycotoxins	13,332	17.5%	**Total**	**76,284**	**100.0%**

**Table 3 ijerph-19-08525-t003:** Results of two-way joining cluster analysis related to notifications on pesticides in the RASFF.

Pesticide (Number of Notifications)		Value (Figure)
Variable		
Acephate (172)	Year	2012–2013
	Notification type	Border rejection (Appendix A)
	Product category	Fruits and vegetables (Appendix A) (okra)
	Origin country	India (Appendix A)
	Notified country	The United Kingdom (Appendix A)
	Notification basis	Border control—consignment detained (Appendix A)
	Distribution status	No distribution (Appendix A)
	Action taken	Destruction (Appendix A)
Acetamiprid (257)	Year	2012–2013, 2020
	Notification type	Border rejection (Appendix A)
	Product category	Fruits and vegetables (Appendix A) (okra—India; peppers, pomegranates—Turkey)
	Origin country	India (2012–2013), Turkey (2020) (Appendix A)
	Notified country	France, the United Kingdom (2012–2013), Bulgaria (2020) (Appendix A)
	Notification basis	Border control—consignment detained (Appendix A)
	Distribution status	No distribution, product not (yet) placed on the market (2012–2013, 2020) (Appendix A)
	Action taken	Destruction (Appendix A)
Carbendazim (333)	Year	2014–2015
	Notification type	Border rejection (Appendix A)
	Product category	Fruits and vegetables (Appendix A)
	Origin country	India (Appendix A)
	Notified country	Italy (Appendix A)
	Notification basis	Border control—consignment detained (Appendix A)
	Distribution status	Product not (yet) placed on the market (2014), product forwarded to destination (2015) (Appendix A)
	Action taken	Destruction (2014–2015), re-dispatch (2015) (Appendix A)
Carbofuran (184)	Year	2014–2019
	Notification type	Information (2014–2017), border rejection (2018–2019), alert (2018) (Appendix A)
	Product category	Fruits and vegetables (Appendix A) (aubergines—The Dominican Republic; goji berries—China)
	Origin country	The Dominican Republic (2014, 2017–2019), Vietnam (2015), Thailand (2016), China (2017–2018) (Appendix A)
	Notified country	Switzerland (2014–2016), the Netherlands (2017–2018) (Appendix A)
	Notification basis	Official control in the market (2016–2018), border control—consignment detained (2018–2019) (Appendix A)
	Distribution status	Product not (yet) placed on the market (2014, 2018–2019), distribution restricted (2015–2016), distribution to other member countries (2017–2018) (Appendix A)
	Action taken	Destruction (2014, 2018–2019), official detention (2016), withdrawal from the market (2017–2018) (Appendix A)
Chlorpyrifos (460)	Year	2015–2020
	Notification type	Border rejection (Appendix A)
	Product category	Fruits and vegetables (Appendix A) (lemons, peppers, stuffed vine leaves)
	Origin country	Turkey (2016–2017) (Appendix A)
	Notified country	Bulgaria (2016–2017) (Appendix A)
	Notification basis	Border control—consignment detained (Appendix A)
	Distribution status	Product not (yet) placed on the market (Appendix A)
	Action taken	Destruction (2016–2019) (Appendix A)
Dichlorvos (112)	Year	2013–2015
	Notification type	Border rejection (Appendix A)
	Product category	Fruits and vegetables (Appendix A) (beans)
	Origin country	Nigeria (Appendix A)
	Notified country	The United Kingdom (Appendix A)
	Notification basis	Border control—consignment detained (Appendix A)
	Distribution status	Product not (yet) placed on the market (Appendix A)
	Action taken	Official detention (2013), re-dispatch (2014), destruction (2014–2015) (Appendix A)
Dimethoate (326)	Year	2012–2013
	Notification type	Border rejection (Appendix A)
	Product category	Fruits and vegetables (Appendix A) (okra—India; oranges—Egypt)
	Origin country	India (2012–2013), Egypt (2013), Kenya (2013) (Appendix A)
	Notified country	The United Kingdom (2012–2013), France (2013) (Appendix A)
	Notification basis	Border control—consignment detained (Appendix A)
	Distribution status	No distribution (2012), product not (yet) placed on the market (2013) (Appendix A)
	Action taken	Destruction (Appendix A)
Ethylene oxide (494)	Year	2020
	Notification type	Alert (Appendix A)
	Product category	Nuts, nut products and seeds (Appendix A) (sesame seeds)
	Origin country	India (Appendix A)
	Notified country	The Netherlands (Appendix A)
	Notification basis	Company’s own check (Appendix A)
	Distribution status	Distribution to other member countries (Appendix A)
	Action taken	Informing consignor, informing recipient(s), withdrawal from market, recall from consumers (Appendix A)
Fipronil (186)	Year	2017
	Notification type	Information (Appendix A)
	Product category	Eggs and egg product (Appendix A)
	Origin country	Italy (Appendix A)
	Notified country	Italy (Appendix A)
	Notification basis	Official control in the market (Appendix A)
	Distribution status	Distribution restricted to notifying country (Appendix A)
	Action taken	Withdrawal from the market (Appendix A)
Formetanate (203)	Year	2012, 2014, 2018–2020
	Notification type	Border rejections (Appendix A)
	Product category	Fruits and vegetables (Appendix A) (peppers)
	Origin country	Turkey (Appendix A)
	Notified country	Bulgaria (Appendix A)
	Notification basis	Border control—consignment detained (Appendix A)
	Distribution status	No distribution (2012), product not (yet) placed on the market (2014, 2018–2020) (Appendix A)
	Action taken	Placed under customs seals (2012), destruction (2014, 2018–2020) (Appendix A)
Methamidophos (176)	Year	2002, 2011–2013
	Notification type	Information (2002), border rejection (2011–2013) (Appendix A)
	Product category	Fruits and vegetables (2002, 2013) (Appendix A) (okra—India)
	Origin country	Turkey (2002), India (2011–2012) (Appendix A)
	Notified country	Germany (2002), France (2012) (Appendix A)
	Notification basis	(not specified) (2002), border control—consignment detained (2012) (Appendix A)
	Distribution status	(not specified) (2002), no distribution (2012) (Appendix A)
	Action taken	(not specified), complaint (2002), destruction (2011–2013) (Appendix A)
Methomyl (260)	Year	2007–2008, 2010–2013, 2018
	Notification type	Information (2007–2008), border rejection (2010–2013, 2018) (Appendix A)
	Product category	Fruits and vegetables (Appendix A) (peppers—Turkey)
	Origin country	Turkey (2010–2011, 2018), the Dominican Republic (2012) (Appendix A)
	Notified country	Bulgaria (2010–2011, 2018), Germany, the Netherlands (2012), France (2013) (Appendix A)
	Notification basis	Official control in the market (2007–2008), border control—consignment detained (2010–2013, 2018) (Appendix A)
	Distribution status	No distribution (2010–2013) (Appendix A)
	Action taken	Destruction (Appendix A)
Monocrotophos (159)	Year	2012–2013
	Notification type	Border rejection (Appendix A)
	Product category	Fruits and vegetables (Appendix A) (okra)
	Origin country	India (Appendix A)
	Notified country	The United Kingdom (Appendix A)
	Notification basis	Border control—consignment detained (Appendix A)
	Distribution status	No distribution (Appendix A)
	Action taken	Destruction (Appendix A)
Omethoate (223)	Year	2005–2013, 2018–2019
	Notification type	Border rejection (2009–2013, 2018–2019), information (2005–2013, 2019) (Appendix A)
	Product category	Fruits and vegetables (2008–2013, 2018–2019) (Appendix A)
	Origin country	Thailand (2008–2010) (Appendix A) (aubergines, beans)
	Notified country	The Netherlands (2008, 2010–2011), Germany (2010), France (2013), the United Kingdom (2013, 2019) (Appendix A)
	Notification basis	Official control in the market (2005–2008, 2010–2011), border control—consignment detained (2009–2013, 2018–2019) (Appendix A)
	Distribution status	No distribution (2008–2012), product already consumed (2011), product not (yet) placed on the market (2013, 2018–2019) (Appendix A)
	Action taken	(not specified) (2007), withdrawal from the market (2009, 2011), destruction (2009–2010, 2013, 2018–2019), inform authorities (2012–2013) (Appendix A)
Oxamyl (160)	Year	2007, 2011
	Notification type	Information (2007), border rejection (2011) (Appendix A)
	Product category	Fruits and vegetables (Appendix A) (peppers)
	Origin country	Turkey (Appendix A)
	Notified country	Germany (2007), Bulgaria (2011) (Appendix A)
	Notification basis	Official control in the market (2007), border control—consignment detained (2011) (Appendix A)
	Distribution status	Product already consumed (2007), no distribution (2011) (Appendix A)
	Action taken	(not specified), no stock left, reinforced checking (2007), destruction, inform authorities, re-dispatch or destruction (2011) (Appendix A)
Profenofos (143)	Year	2012–2013
	Notification type	Border rejection (Appendix A)
	Product category	Herbs and spices (2012), fruits and vegetables (2013) (Appendix A) (curry, okra)
	Origin country	India (Appendix A)
	Notified country	France (Appendix A)
	Notification basis	Border control—consignment detained (Appendix A)
	Distribution status	No distribution (Appendix A)
	Action taken	Destruction (Appendix A)
Triazophos (213)	Year	2012–2013
	Notification type	Border rejection (Appendix A)
	Product category	Herbs and spices (2012), fruits and vegetables (2012–2013) (Appendix A) (curry, okra)
	Origin country	India (Appendix A)
	Notified country	France (2012), the United Kingdom (2012–2013) (Appendix A)
	Notification basis	Border control—consignment detained (Appendix A)
	Distribution status	No distribution (Appendix A)
	Action taken	Destruction (Appendix A)

**Table 4 ijerph-19-08525-t004:** Information on notifications of pesticide residues in the RASFF according to different authors.

Period	Product (Product Category)	Pesticide(s)	Origin Country	Notifying Country	Reference
2002–2015	Food of non-animal and animal origin	NDA	NDA	NDA	[34]
2002–2019	Gherkins	Oxamyl	Turkey	NDA	[35]
2003–2007	Fruits and vegetables	Dimethoate, isofenphos-methyl, omethoate, oxamyl, methamidophos, methomyl, monocrotophos	NDA	NDA	[36]
2003–2013	Grapes	Carbendazim, methomyl, oxamyl	NDA	NDA	[37]
2004–2014	Herbs and spices	Chlorpyrifos, triazophos	NDA	NDA	[38]
2005–2015	Fruits and vegetables	NDA	NDA	NDA	[39]
2006–2013	Baby or infant products	NDA	The European Union	NDA	[40]
2008–2011	Fruits and vegetables, herbs and spices	NDA	NDA	NDA	[41]
2008–2011	Fruits and vegetables, herbs and spices, nuts and nut products	NDA	NDA	NDA	[42]
2008–2013	Fruits and vegetables	NDA	Africa	NDA	[43]
2009–2020	Rice	NDA	Pakistan	NDA	[44]
before 2011	Okra, curry leaves	NDA	India	NDA	[32]
2012	Fresh pepper	NDA	Turkey	NDA	[45]
2012	Fruits and vegetables	Monocrotophos	India	NDA	[46]
2013	Fruits and vegetables	NDA	NDA	NDA	[47]
2013–2014	Fresh mint	Carbendazim	NDA	NDA	[48]
2013–2015	Fruits and vegetables, spices	NDA	India	NDA	[49]
before 2014	Okra, curry leaves	NDA	India	NDA	[4]
2014	Fruits and vegetables	Dichlorvos	Nigeria	United Kingdom	[50]
2014–2018	Herbs and spices	Chlorpyrifos	NDA	NDA	[51]
2015–2015	Cumin	NDA	India	NDA	[52]
2015–2020	Fruits and vegetables	NDA	NDA	NDA	[53]
2016	Fruits and vegetables	NDA	Turkey	Bulgaria, the Netherlands	[54]
2016	Black and green teas	Propargite	NDA	NDA	[28]
2017	Eggs	Fipronil	Belgium	Belgium	[55]
2017	Eggs and egg products	Fipronil	Italy	Italy	[56]
2019	Fruits and vegetables	Chlorpyrifos	NDA	NDA	[57]
before 2020	Okra	NDA	India	NDA	[58]
2020	Fruits and vegetables	NDA	Turkey	Bulgaria	[59]
2020	Sesame seeds	Ethylene oxide	India	Belgium	[60]
2020	Sesame seeds	Ethylene oxide	India	The Netherlands	[61]
2020	Nuts, nut products and seeds	Ethylene oxide	India	Germany, The Netherlands	[59]

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
