# Peer review of "Notifications on Pesticide Residues in the Rapid Alert System for Food and Feed (RASFF)"

_ijerph, 2022, doi:10.3390/ijerph19148525_

Round 1
Reviewer 1 Report
The manuscript represents a meta-analysis of notifications in the RASFF. As such, the manuscript is a valuable overview of real problems regarding pesticide residues and possibly a good starting point for future strategies and policies. The manuscript is clear, relevant to the field, and presented in a well-structured manner. Material and methods are sound, and results are well defined with self-explanatory tables.
Author Response
I would like to thank the Reviewer for positive review of the proposed paper. Regarding future strategies and policies, in the Conclusions I have expanded the paragraph on training on the use of pesticides in non-EU countries, which should be carried out at the level of working teams, set up jointly by the EU institutions and the authorities and bodies of individual countries.
Reviewer 2 Report
I enjoyed reading this paper.
The paper deals with an interesting and quite relevant topic as pesticide residues on food and highlights the efficiency of the European control system.
However, some aspects could be improved in my opinion.
The introduction is a bit confusing.
Lines 30 to 37 - I suggest referring to EFSA documents to better explain the definition of MRL and why this parameter is important
Lines 39 to 44 the text is a bit of a repetition on why pesticides are used, I suggest eliminating it. Reference document 6 is interesting but should be better contextualized in the definition of MRL and residues in general
Lines 50 and 51 - this text should be placed at the beginning.(It is estimated that if pesticides were not used, a third of crops would be lost [2]. However, if pesticides are used in the improper way, they can also be harmful to people, animals and the environment [1].)
lines 52 and 57 - these statements are important and should be supported by a mass of data and literature which are however conflicting. They are also not supported by a dietary risk assessment based on diet and intake. Please remove it, or better justify it.
A more general point: I found the layout a bit dense. Author might consider a few more sub-headings that would help break it up especially the results section.
I appreciated the conclusions in particular the part concerning the need to increase the awareness of pesticide users through training.This section in my opinion, should be highlighted as a stimulus of the European control system to improve the quality production systems in particular in some non-European countries
Author Response
I would like to thank the Reviewer for valuable comments on the proposed paper. They are included below in italic with my response to them and the changes made (plain font).
I enjoyed reading this paper.
The paper deals with an interesting and quite relevant topic as pesticide residues on food and highlights the efficiency of the European control system.
However, some aspects could be improved in my opinion.
The introduction is a bit confusing.
Lines 30 to 37 - I suggest referring to EFSA documents to better explain the definition of MRL and why this parameter is important
The following paragraph was added:
“Under Regulation (EC) 396/2005 the European Food Safety Authority (EFSA) provides annual reports, which examines pesticide residue levels in food on the European Union (EU) market. The main results of these reports from last five years available (2016-2020) were shown in Table 1. These results are based on the analysis of an average of around 90,000 product samples from both EU and non-EU countries each year.”
In new added Table 1 presented the following columns: Year, Within the legal limits, Below the LOQ (i.e. Limit of Quantification), Not exceeding MRLs, Exceeding MRLs.
Below Table 1 added the sentences:
“The results published by the EFSA indicate that about 95% of the samples are within the legal limits, of which for about 40% limits of quantification (LOQ) were not even exceeded, and only in 2-3% of the samples (taking into account the measurement uncertainty) the MRLs were exceeded. In the aforementioned reports the EFSA points out that the probability of European citizens being exposed to levels of pesticide residues that could lead to adverse health effects is low. However, it consistently recommends the need to improve the efficiency of the European control system for pesticide residues [6-10], including optimising traceability.”
Referring to the Reviewer’s next comment (below), the MRL issue has been further expanded.
Lines 39 to 44 the text is a bit of a repetition on why pesticides are used, I suggest eliminating it. Reference document 6 is interesting but should be better contextualized in the definition of MRL and residues in general.
Lines 40-44 have been removed. A new paragraph has been created. To the sentence “Besides, climatic conditions in the some countries can promote plant diseases (caused by plant pathogens), plant pests (plant-feeding or -sucking insects and mites) and weeds and thus reduce yields” was added: “In this context, it is therefore very important to apply proper good agricultural practice (GAP), which means the nationally recommended, authorised or registered safe use of plant protection products under actual conditions at any stage of production, storage, transport, distribution and processing of food. In addition, the principles of integrated pest control in a given climate zone should be implemented, as well as the use of a minimum quantity of pesticides and the setting of MRLs/temporary MRLs at the lowest level that achieves the intended effects. However, in certain cases it is necessary to implement a critical GAP, which means more than one GAP for an active sub-stance/product combination leading to an increase in the highest acceptable level of pesticide residues in treated crops and the basis for establishing MRL”.
In addition, in the following paragraph, the data on active substances have been updated and the following sentence has been added: “This database also includes information on pesticide residues and the MRLs that apply for such residues in food products – 654 records (as of 5.07.2022)”.
Lines 50 and 51 - this text should be placed at the beginning.(It is estimated that if pesticides were not used, a third of crops would be lost [2]. However, if pesticides are used in the improper way, they can also be harmful to people, animals and the environment [1].)
This passage of text has been moved to the first paragraph of the paper (at the end of this paragraph).
lines 52 and 57 - these statements are important and should be supported by a mass of data and literature which are however conflicting. They are also not supported by a dietary risk assessment based on diet and intake. Please remove it, or better justify it.
Lines 51-58 have been removed.
A more general point: I found the layout a bit dense. Author might consider a few more sub-headings that would help break it up especially the results section.
Section “2. Materials and methods” was divided into two subsections: “2.1. Data and its processing” and “2.2. The cluster analysis”
In section “3. Results” subsection “3.1. General results” was divided into two further subsections: “3.1.1. Percentage share of notifications” and “3.1.2. Results of joining cluster analysis”, and the name of subsection 3.2 was modified to “Results of two way-joining cluster analysis for pesticides studied” and it was also divided into five further subsections: “3.2.1. Fruits, vegetables and nuts from India”, “3.2.2. Fruits and vegetables from Turkey”, “3.2.3. Vegetables from Nigeria”, “3.2.4. Eggs from Italy” and “3.2.5. Other notifications”
I appreciated the conclusions in particular the part concerning the need to increase the awareness of pesticide users through training. This section in my opinion, should be highlighted as a stimulus of the European control system to improve the quality production systems in particular in some non-European countries.
The last paragraph of the Conclusions has been divided, rebuilt and widened:
“It is necessary to develop cooperation between EU institutions and bodies and their counterparts in non-EU countries. Cooperation could take place at the level of working teams and include advising on the creation or amendment of laws, as well as providing training in these countries. Training should take into account the indigenous or even local nature of production in these countries, be conducted in a structured manner and with the participation of national supervisory authorities and bodies, and cover issues related to the use of pesticides permitted by the EU and appropriate for a given plant (season, amount, withdrawal period). This will minimise or avoid image and financial losses (e.g. costs related to transport and disposal of contaminated products) by countries exporting products to the EU market.”
Reviewer 3 Report
This paper analyzes the pesticide detection of RASFF, which is very important for food safety issues. The following are my comments on this paper.
1. The research method of this paper is not very special, so we can only do simple statistics and description of test results. In this paper, only the quantity of pesticide residues is indicated, and the research conclusion lacks major academic findings.
2. This paper only explains the pesticide residues part of RASFF. I suggest that the authors should indicate the gap between the allowable standard and the excess standard of pesticide residues, so as to show the severity of pesticide residues.
3. I suggest that the "5. Conclusions" of this paper should emphasize whether it is different from other research results.
Author Response
I would like to thank the Reviewer for valuable comments on the proposed paper. They are included below in italic with my response to them (plain font).
This paper analyzes the pesticide detection of RASFF, which is very important for food safety issues. The following are my comments on this paper.
- The research method of this paper is not very special, so we can only do simple statistics and description of test results. In this paper, only the quantity of pesticide residues is indicated, and the research conclusion lacks major academic findings.
Indeed, two-way joining cluster analysis is not often used, but I have applied it previously to RASFF notification studies in already published papers on heavy metals (in 2018), pathogenic microorganisms (2018), both in this Journal and mycotoxins (2019), in “Quality Assurance and Safety of Crops & Foods”. The proposed paper using this method thus completes the series of articles on the most frequently reported hazards in the RASFF. It allowed the aggregation of a large amount of data and highlights the similarities that occur.
After the introduction of corrections, in Conclusions the paragraph on trainings in the use of pesticides in non-EU countries was separated and expanded. A proposal to conduct further research has also been added:
„Further research on pesticide residues could include linking RASFF notifications to quantities of particular types of imported products (based on Eurostat data) in order to forecast possible hazards. However, this could be hampered by data gaps in the RASFF database (especially for the earlier years of the system operation) and the lack of access to historical data in the currently officially functioning database. Therefore, it would be necessary to additionally connect the data from the historical database and the one currently available.”
- This paper only explains the pesticide residues part of RASFF. I suggest that the authors should indicate the gap between the allowable standard and the excess standard of pesticide residues, so as to show the severity of pesticide residues.
The purpose of the paper related to the RASFF notifications and the analysis and conclusions were based on them. However, the section Introduction has now been significantly expanded, compiling the main results published in the annual reports on pesticide residues issued by the European Food Safety Authority for last available five years (1996-2020): within the legal limits, below limit of quantification (LOQ), not exceeding maximum residue levels (MRLs) and exceeding MRLs. Further in the Introduction, a definition of good agricultural practice (GAP) and critical GAP have also been added in the context of pesticide residues.
- I suggest that the "5. Conclusions" of this paper should emphasize whether it is different from other research results.
The following passage has been added to the Conclusions:
“The detailed study covered half of the RASFF notifications for the seventeen most frequently notified pesticides between 1994 and 2020, i.e. a 27-year period. The two-way joining cluster analysis enabled the identification of concentrations and similarities in the notifications indicating, among others: pesticide, years, product, origin country and notifying country.”
Round 2
Reviewer 3 Report
Although the authors have made some explanations, they still cannot fully explain these three concerns, the authors also did his best to make some corrections. As a result, the paper can now be accepted for publication as is.